# Integration of Sentinel-1A, ALOS-2 and GF-1 Datasets for Identifying Landslides in the Three Parallel Rivers Region, China

Cong Zhao [1], Jingtao Liang [1,*], Su Zhang [1], Jihong Dong [1,2], Shengwu Yan [1], Lei Yang [1], Bin Liu [1], Xiaobo Ma [1] and Weile Li [3]

1 Evaluation and Utilization of Strategic Rare Metals and Rare Earth Resource Key Laboratory of Sichuan Province, Sichuan Geological Survey, Chengdu 610081, China
2 Sichuan Intelligent Geological Big Data Co., Ltd., Chengdu 610081, China
3 State Key Laboratory of Geohazard Prevention and Geoenvironment Protection, Chengdu University of Technology, Chengdu 610059, China
* Correspondence: liangjingtao@scsdzdcy.onaliyun.com

**Abstract:** In the process of using InSAR technology to identify active landslides, phenomena such as steep terrain, dense vegetation, and complex clouds may lead to the missed identification of some landslides. In this paper, an active landslide identification method combining InSAR technology and optical satellite remote sensing technology is proposed, and the method is successfully applied to the Three Parallel Rivers Region (TPRR) in the northwest of Yunnan Province, China. The results show that there are 442 landslides identified in the TPRR, and the fault zone is one of the important factors affecting the distribution of landslides in this region. In addition, 70% of the active landslides are distributed within 1 km on both sides of the fault zone. The larger the scale of the landslide, the closer the relationship between landslides and the fault zone. In this identification method, the overall landslide identification rate based on InSAR technology is 51.36%. The combination of Sentinel-1 and ALOS-2 data is beneficial to improve the active landslide identification rate of InSAR. In the northern region with large undulating terrain, shadows and overlaps occur easily. The southern area with gentle terrain is prone to the phenomenon where the scale of landslides is too small. Both phenomena are not conducive to the application of InSAR. Thus, in the central region, with moderate terrain and slope, the identification rate of active landslides based on InSAR is highest. The active landslide identification method proposed in this paper, which combines InSAR and optical satellite remote sensing technology, can integrate the respective advantages of the two technical methods, complement each other's limitations and deficiencies, reduce the missed identification of landslides, and improve the accuracy of active landslide inventory maps.

**Keywords:** Three Parallel Rivers Region (TPRR); landslides; InSAR; optical satellite; fault zone

## 1. Introduction

The Three Parallel Rivers Region (TPRR) is located in the Hengduan Mountains region in the northwest of Yunnan Province, China, and it is the intersection of the three geographic regions of the Qinghai–Tibet Plateau, East Asia, and South Asia [1]. The Jinsha River, Nujiang River, and Lancang River are three well-known rivers in Asia, flowing in parallel from north to south in this area, forming three deep canyons.

The TPRR is characterized by its beautiful geological landforms, rich biodiversity, and spectacular natural landscapes, and has been included in the "World Natural Heritage List" by UNESCO [2–4]. The fragile geological environment and frequent geological tectonic movements in the TPRR, coupled with the impact of human engineering activities, such as the development of a series of cascade hydropower stations built in the Jinsha River and Lancang River [5,6], make this area one of the most developed landslide geohazard

areas in China [7,8]. Therefore, it is particularly important to carry out research work on the distribution, activity, and evolution mechanisms of landslides in the TPRR.

Creating a regional landslide inventory map is the core focus of landslide investigation and research [9,10]. The early landslide inventory maps mainly relied on the traditional field geological surveys [11,12]. Nowadays, optical satellite image remote sensing [13,14], UAV aerial images [15], airborne LiDAR remote sensing [16–18], InSAR monitoring [19–27], and other advanced technical methods are used to create landslide inventory maps. Among them, optical remote sensing has become a widely used technology in landslide identification and investigation due to it being intuitive and convenient, and InSAR has gradually become an indispensable and important technical method in landslide investigation and research due to its advantages of wide coverage, long monitoring periods, and accurate surface deformation information [21].

At present, many scholars have carried out InSAR identification and monitoring of landslide geohazards in the TPRR. For example, Zhao et al. used ALOS/PALSAR-1 to identify potential landslides in the Wudongde reservoir area of the Jinsha River [23]; Liu et al. integrated Sentinel-1 and ALOS/PALSAR-2 SAR datasets to create a survey map of active landslides in the Jinsha River basin [24]; Wang et al. created a landslide inventory map in the lower Niulan River, a tributary of the Jinsha River, based on Sentinel-1 and ALOS/PALSAR SAR datasets [25]; Cao et al. compared the identification effectivity of potential landslides in the lower Jinsha River from Sentinel-1 data and ALOS-2 data [26]. The above studies are mostly concentrated in the Jinsha River Basin, where hydropower development is more intensive, and the identification and investigation of landslides have not been widely carried out in the Lancang River Basin and the Nu River Basin. In addition, when only using InSAR technology to carry out landslide identification and investigation, the high density of vegetation in mountainous areas and the excessive terrain height difference often lead to the failure of InSAR technology in some areas.

Based on the Sentinel-1A and ALOS-2 datasets, combined with the GF-1 optical satellite, in this study, we carried out landslide investigation and research work within a total area of $6.58 \times 10^4$ km$^2$ in the TPRR, in the northwest of Yunnan Province, China, and created a landslide inventory map in this area; we then analyzed the distribution and activity characteristics of landslides in the area. This paper proposes an active landslide identification method combining InSAR technology and optical satellite remote sensing, discusses the application effect of this method in the TPRR, compares and analyzes the applicability of Sentinel-1A data and ALOS-2 data in this area, and summarizes the impact of different factors in terms of landslide identification. The research results are beneficial to landslide prevention and early warning in the TPRR and provides a reference to establish a mature method of active landslide identification that combines InSAR technology and optical satellite remote sensing technology in a large area.

## 2. Study Area

The Three Parallel Rivers Region (TPRR) is located in the Hengduan Mountains, on the southeastern edge of the Qinghai–Tibet Plateau. The area is lined with high mountains and deep canyons, and the terrain has a large difference in altitude. The overall terrain is high in the north and low in the south, with a step-like decline.

Well-known large mountains and rivers, such as the Dulong River, Gaoligong Mountain, Nujiang River, Nushan Mountain, Lancang River, Yunling River, and Jinsha River, are arranged vertically from west to east, forming the main area of the TPRR (Figure 1). The Three Parallel Rivers Region is the main research area of this paper, including 20 counties (cities) of the Dali Bai Autonomous Prefecture, Diqing Tibetan Autonomous Prefecture, Nujiang Lisu Autonomous Prefecture, and Lijiang City, with a total area of approximately $6.58 \times 10^4$ km$^2$.

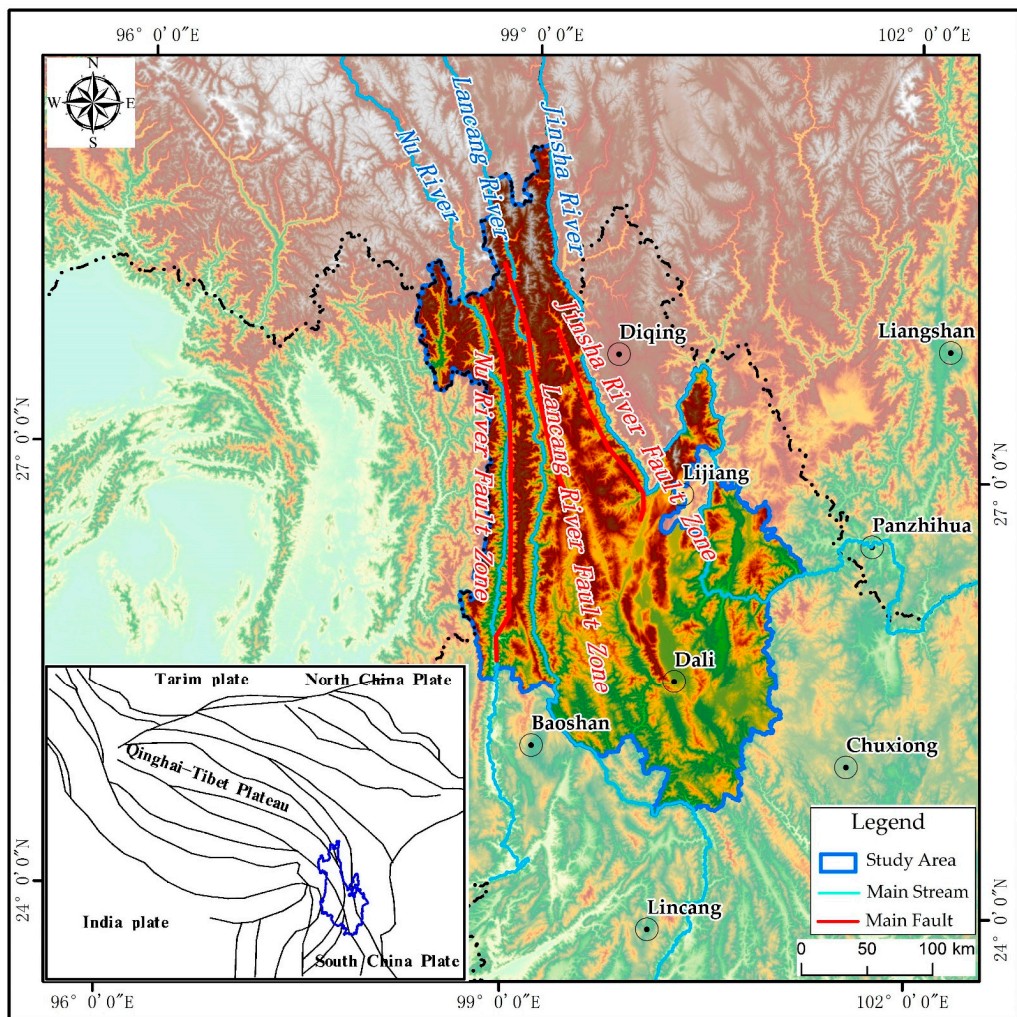

**Figure 1.** Location of the study area.

The TPRR is located in the collision and suture zone of the Eurasian plate and the Indian plate. The geological structure is extremely complex, and the tectonic lines are distributed in the north–south direction.

The combination of three deep and large fault zones—the Nujiang river fault zone, the Lancang river fault zone, and the Jinsha river fault zone—plays a key controlling role in the geological conditions of the study area [8]. The fragile geological conditions lead to the strong development of landslide geohazards in the area. In addition, the TPRR is very rich in water resources, and many large-scale hydropower stations have been built in the Jinsha River and Lancang River basins. The development of the hydropower stations further induces the development of landslide geohazards in the area [24].

Due to the large area of the TPRR, the topography and landforms of each region are quite different, and the landslide scale and typical deformation characteristics of each region are significantly different. To facilitate the subsequent research and analysis, according to the difference in altitude and the intensity of terrain cutting, the study area is divided into three areas: the north area, the central area, and the south area (Figure 2).

The northern region is a typical alpine and canyon landform. According to DEM calculations, the average elevation of this region is 3140 m, the average terrain slope is 27°, and the total area is about 25,100 km². The central region is a typical middle-alpine landform, the elevation is 2470 m above sea level, the average terrain slope is 21°, and the total area is about 23,900 km². The southern region is a typical low-mountain hilly landform with an average elevation of 2080 m and an average terrain slope of 17°, and the area is about 16,800 km².

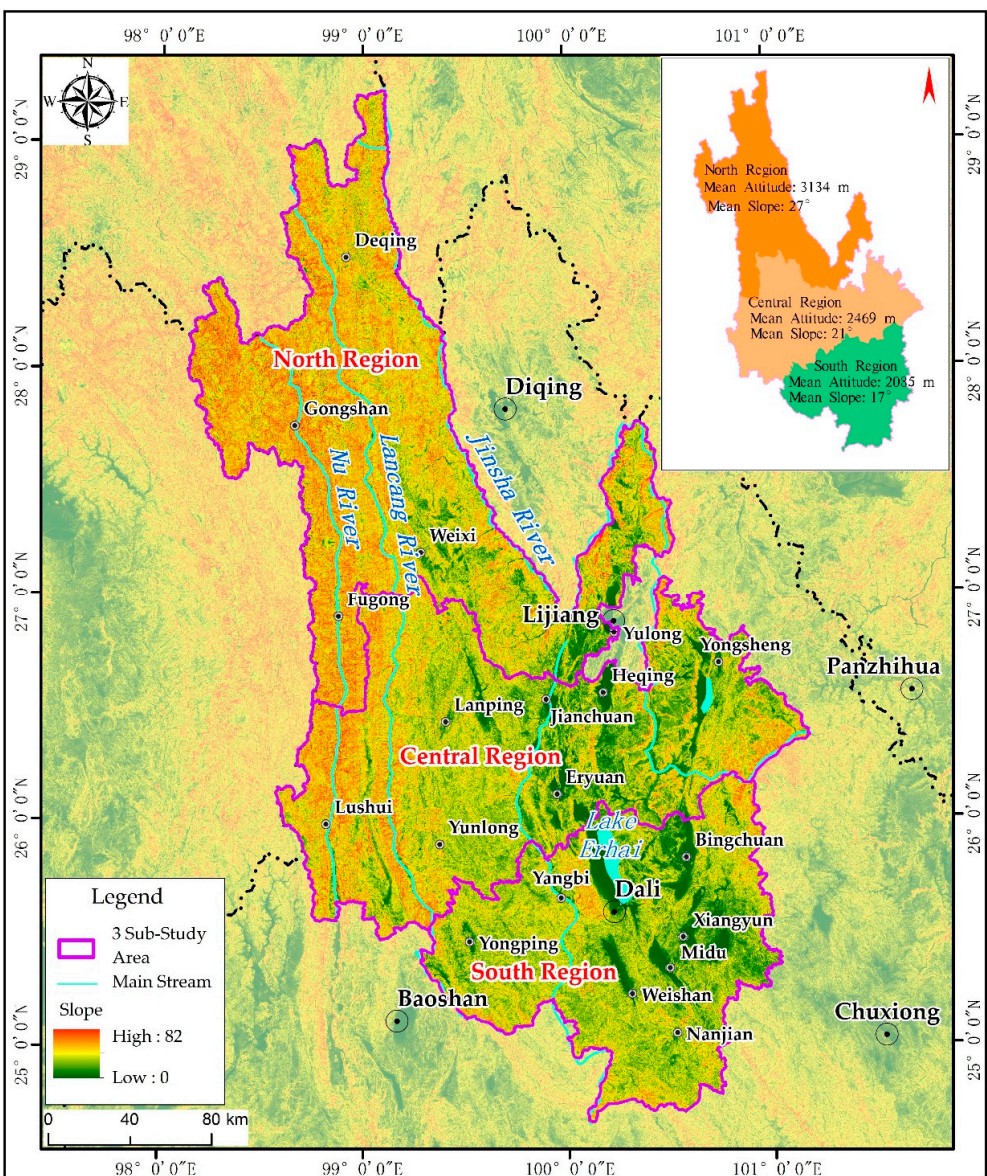

**Figure 2.** Slope map and regional division map of the study area.

## 3. Materials and Methods

### 3.1. Data Source

3.1.1. SAR Satellite Dataset

The SAR satellite data used in this paper include Sentinel-1A data and ALOS-2 data (Figure 3). The Sentinel-1A satellite was developed by the European Space Agency (ESA) and started operation in October 2014. The Sentinel-1A satellite is a C-band synthetic aperture satellite with a return period of 12 days and a total of 4 imaging modes: interferometric wide mode, wave mode, strip mode, and ultra-wide mode. This study adopts the interferometric wide (IW) mode of Sentinel-1A satellite data, the polarization mode is VV, the ground resolution is 5 × 20 m, the orbit type includes the ascending orbit and descending orbit, and the time span is from January 2017 to January 2020. The Sentinel-1A satellite data cover the whole region, with a total of 70 to 80 periods. The Sentinel-1A SAR satellite data information is shown in Table 1.

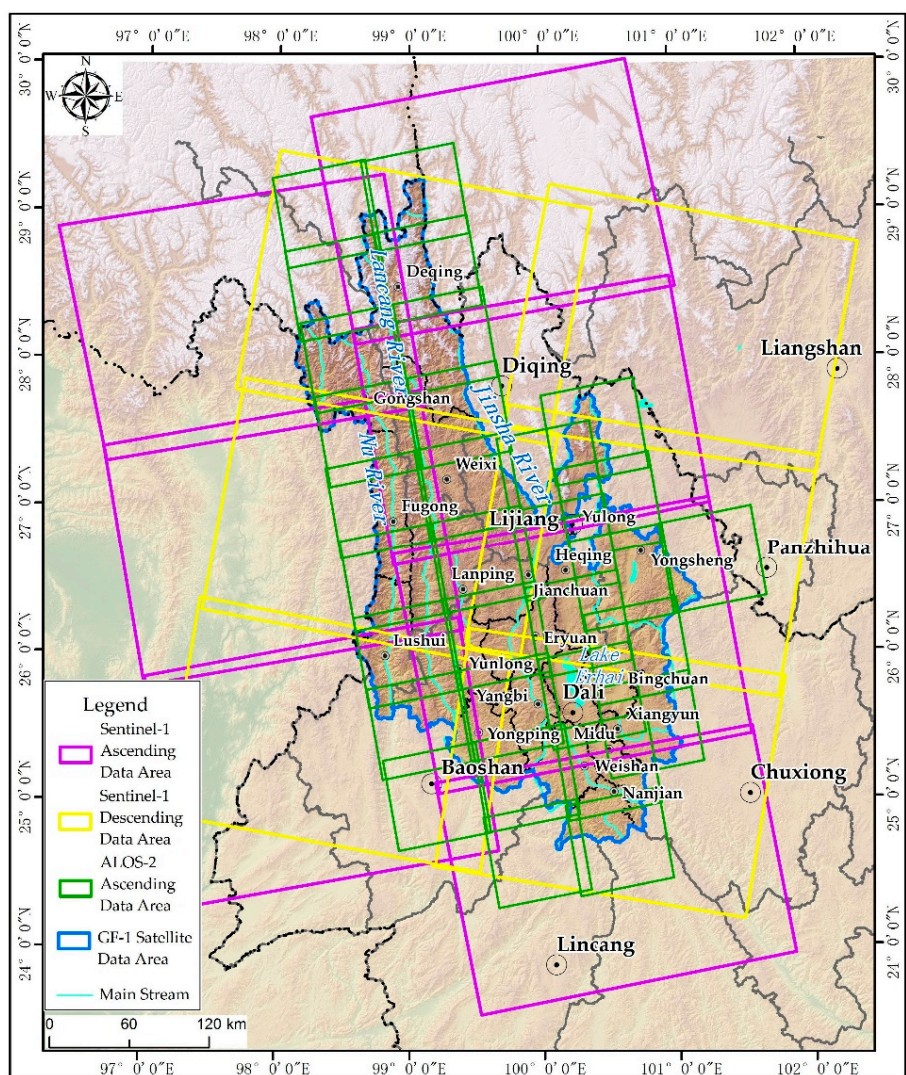

**Figure 3.** Coverage of the Sentinel-1A and ALOS-2 datasets.

**Table 1.** Main parameters of SAR satellite data.

| Satellite | Orbital Direction | Band | Wavelength | Resolution | Temporal Coverage |
|---|---|---|---|---|---|
| Sentinel-1A | Ascending, Descending | C-band | 5.6 cm | $5 \times 20$ m | 2017.01–2020.01 |
| ALOS-2 | Ascending | L-band | 23.5 cm | $4 \times 4$ m | 2018.07–2019.07 |

The ALOS-2 satellite was launched by the Japan Aerospace Exploration Agency (JAXA) in 2014. This satellite is currently the most commonly used L-band synthetic aperture satellite in the world [24–27]. It has the characteristic of strong penetration. It has advantages in areas with lush vegetation coverage, large surface fluctuations, and a humid climate, and it is easier to form interference image pairs. This study uses the ascending orbit data of the ALOS-2 satellite, the polarization mode is VV, and the ground resolution is $4 \times 4$ m. The time span is from July 2018 to July 2019, and the ALOS-2 satellite covers the entire region. Each scene covers from 5 to 8 periods. The ALOS-2 satellite data information is shown in Table 1.

The digital elevation models (DEMs) used in the InSAR data processing are all from the shuttle radar topography mission (SRTM), with a spatial resolution of 30 m [28].

### 3.1.2. Optical Satellite Data

In this study, the GF-1 optical satellite image was also used for the identification of active landslides in the TPRR. The spatial resolution of the panchromatic band and multispectral band is 2 and 8 m, and the imaging time is 2019–2020.

### 3.2. *Data Processing Method*

### 3.2.1. InSAR Data Processing Method

Stacking-InSAR technology is mostly used for the identification of active landslides [29,30], while SBAS-InSAR technology can be used not only for the identification of active landslides, but also for the analysis of the deformation process of landslides [31,32]. The key content of this study is the identification of active landslides in the TPRR, and the specific analysis of the landslide deformation process is not within the scope of this study. Therefore, here, both Sentinel-1A data and ALOS-2 data were processed with Stacking-InSAR.

Stacking-InSAR technology is based on D-InSAR and obtains the surface deformation rate result by performing a weighted average solution on multiple differential interferograms [33]. The data processing flow of Stacking-InSAR is shown in Figure 4. The premise of Stacking technology processing is to assume that the surface deformation trend is linear deformation. The phase stacking can effectively suppress atmospheric phase and DEM errors, and improve the relative accuracy of deformation information. The advantage of this technology is that by processing a relatively small amount of SAR data, it can quickly obtain the distribution of landslide geohazards in a large area, and the reliability of the results is relatively high [21,22].

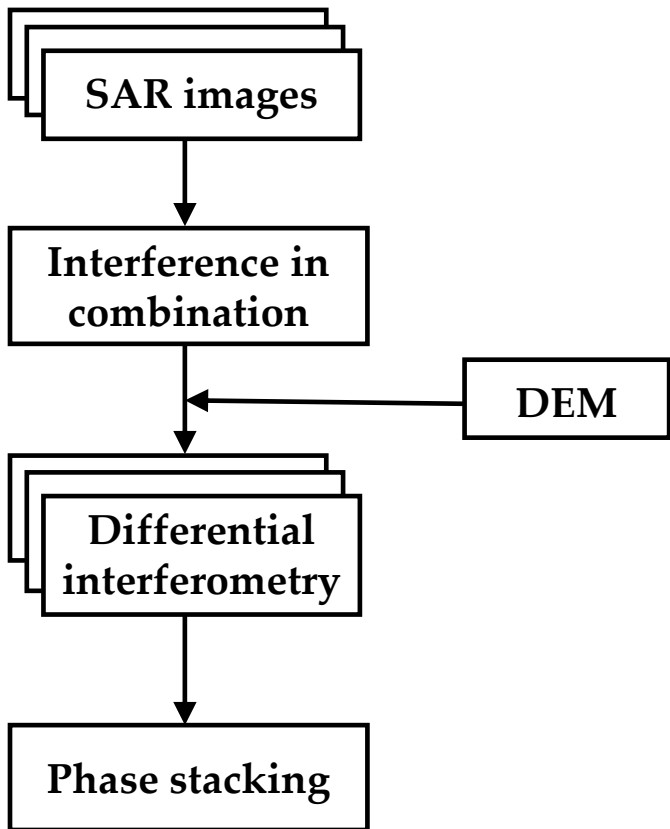

**Figure 4.** Stacking-InSAR data processing flow.

### 3.2.2. Identification Method of Active Landslides

In this study, an active landslide identification method combining InSAR technology and optical satellite remote sensing technology is used. The identification process is shown in Figure 5.

First, the method processes the SAR data and optical satellite image raw data of the study area, respectively, to obtain the surface deformation map and optical satellite image of the study area. On this basis, the active landslides in the study area are identified by the following three methods. (1) The surface deformation rate of this area is abnormal. Based on the optical satellite image, it is judged that the area has the terrain conditions for landslide development, but there are no obvious optical deformation characteristics; then, this area can be identified as an active landslide (Figure 5a). (2) The surface deformation rate of this area is abnormal. Based on the optical satellite image, it is judged that the area has the terrain conditions for landslide development and has obvious optical deformation characteristics; then, the area can be identified as an active landslide (Figure 5b). (3) There is no abnormal surface deformation rate in this area, but, based on the optical satellite image, the area has the terrain conditions for landslide development and has obvious optical deformation characteristics; thus, the area can be identified as an active landslide (Figure 5c).

The obvious optical deformation characteristics of active landslides refer to the chair-like shape of the landslide, the local sliding characteristics with obvious color difference, as well as large-scale cracks, staggered platform ridges, etc. This study uses the manual search approach to identify the optical deformation characteristics of landslides.

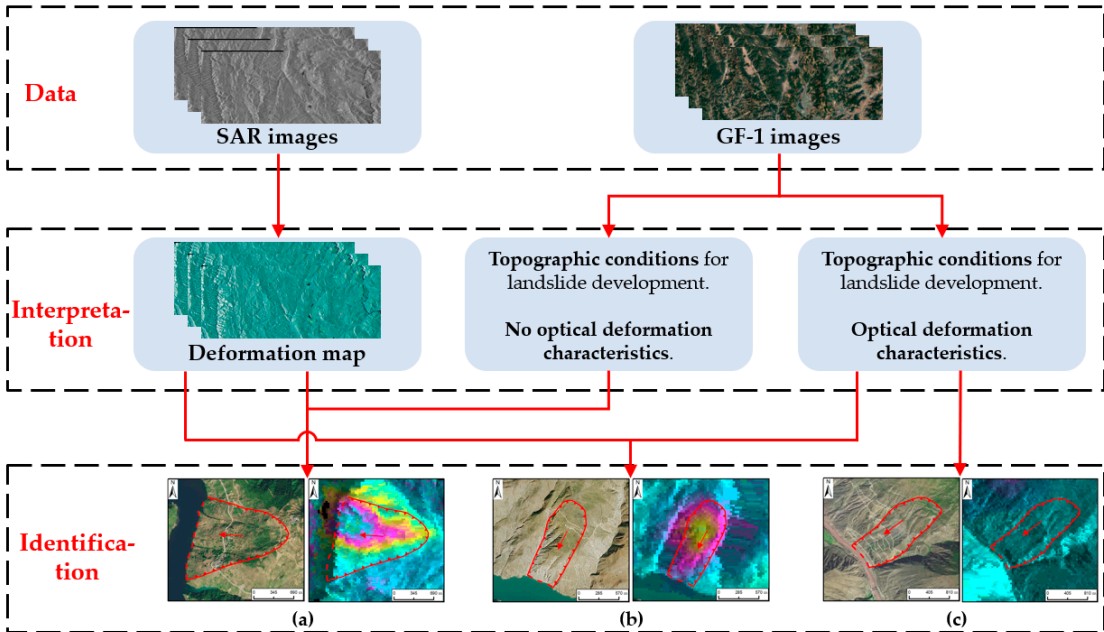

**Figure 5.** Flowchart of active landslide identification ((**a**) active landslide with abnormal surface deformation and no obvious optical deformation characteristics; (**b**) active landslide with abnormal surface deformation and obvious optical deformation characteristics; (**c**) active landslide with obvious optical deformation characteristics without abnormal surface deformation).

## 4. Results

In this study, the landslide identification method combining the above-mentioned InSAR technology and optical satellite remote sensing technology was used to identify a total of 442 active landslides in the TPRR, including 162 active landslides in the northern area, 183 active landslides in the central area, and 97 active landslides in the southern district (Figure 6). Active landslides in the TPRR are mainly distributed along the Lancang River and Jinsha River, and sporadic active landslides can be seen in other areas. In the Lancang River basin and Jinsha River basin, the distribution density of active landslides in the reservoir area of the hydropower station is much higher than that in other areas, which shows that the hydropower stations are important factors affecting the development of active landslides, which is consistent with the research results of other scholars [22–26]. It

can be found from Figure 6 that the distribution characteristics of active landslides in the study area are closely related to the three deep and large faults in the TPRR, and a large number of active landslides are intensively developed around the three major faults and their secondary faults. In the Section 5, we further analyze the relationship between the distribution of active landslides and fault structures in the TPRR.

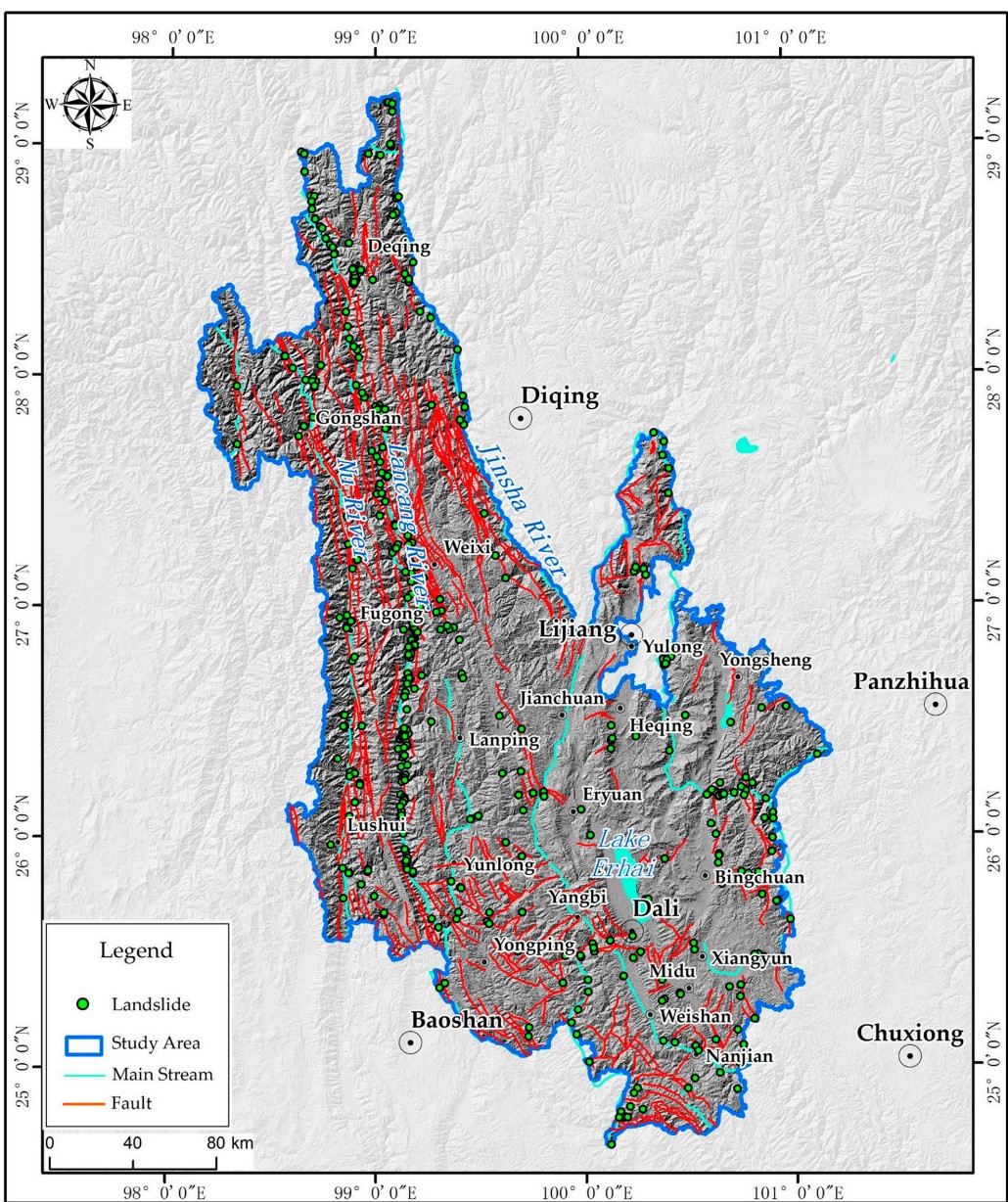

**Figure 6.** Distribution map of active landslides in the TPRR.

## 5. Discussion

### 5.1. Landslide Terrain and InSAR Identification Effect

The elevation and slope of the mountain are important factors affecting the formation and development of landslides [34]. The landslides identified in the TPRR are divided into three gradient intervals according to 0–15°, 16–30°, and >30°. Then, the distribution characteristics of active landslides in the northern, central, and southern areas are summarized. In the northern area, with an average altitude of 3140 m, landslides are mainly distributed in the slope range of 16–30° (86 landslides) and in the slope range of >30° (71 landslides). In the central region, with an average altitude of 2470 m, the landslides are mainly distributed in the slope range of 16–30° (112 landslides), followed by the slope

range of >30° (53 landslides). In the southern area, with an average altitude of 2080 m, the landslides are mainly distributed in the slope range of 16–30° (69 landslides), followed by the slope range of 0–15° (16 landslides). Therefore, it can be seen that, in the TPRR, the degree of cut between the topography and slope of the mountain gradually decreases from north to south, and the slope of the landslides also shows an overall downward trend.

The contribution rate of InSAR technology in the active landslide identification method was further analyzed, and the number of active landslides identified by the three identification methods proposed in the previous article was counted: (1) the number of active landslides with abnormal InSAR surface deformation rates, and with the terrain conditions for landslide development and no obvious optical deformation characteristics, is taken as $n_1$; (2) if the InSAR surface deformation rate is abnormal, with the terrain conditions for landslide development, and the landslides have obvious optical deformation characteristics, the number of such landslides is taken as $n_2$; (3) if the active landslides have no abnormal InSAR surface deformation rates, with the terrain conditions for landslide development, and have obvious optical deformation characteristics, the number is taken as $n_3$.

Therefore, the contribution of InSAR technology in the active landslide identification method proposed in this paper, i.e., the active landslide identification efficiency $\eta_{InSAR}$ based on InSAR technology, can be expressed as:

$$\eta_{InSAR} = \frac{n_1 + n_2}{n_1 + n_2 + n_3} \tag{1}$$

According to statistics, the $\eta_{InSAR}$ in the northern area is 38.89%, the $\eta_{InSAR}$ in the central area is 66.12%, and the $\eta_{InSAR}$ in the southern area is 44.33%. The relationship between the efficiency of active landslide identification by InSAR and altitude as well as slope in the three regions is shown in Figure 7.

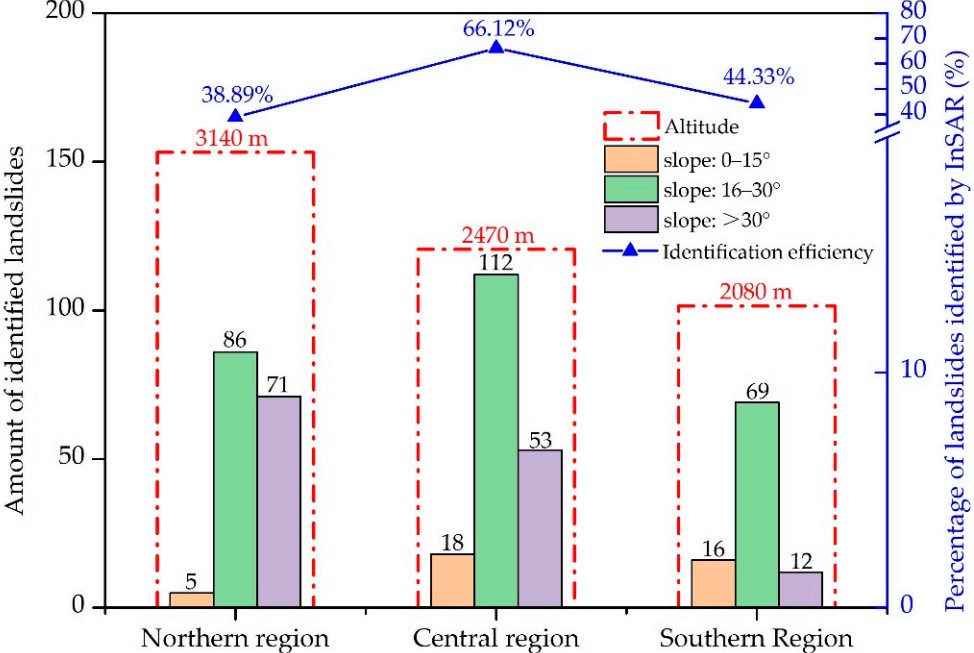

**Figure 7.** Comparison of InSAR identification efficiency of active landslides.

From the northern region to the southern region, the overall altitude of the mountain and the slope of the active landslide show a downward trend. However, the identification efficiency of the active landslides based on InSAR technology increases first and then decreases. In the central area, the identification efficiency of active landslides based on InSAR technology is highest.

Taking the landslide No. GSHP005 in Gongshan County, in the norther region, as an example, we specifically analyzed the reasons for the low identification rate of active landslides based on InSAR technology in the northern region. The landslide is located on the slope of the east bank of the Dimaluo River. The maximum altitude difference in this area is around 2000 m, and the terrain slope is around 35–40°, which belongs to a typical alpine and canyon landform. From the optical satellite image, the left side of the slope shows signs of recent secondary sliding, the sliding body is gray–white, and the overall deformation characteristics of the landslide are obvious (Figure 8a). However, due to the steep terrain and dense vegetation on the landslide, the InSAR result of Sentinel-1A data shows no obvious deformation zone (Figure 8b), and the InSAR result of ALOS-2 data is obtained in the layover shadowed area (Figure 8c). Therefore, the landslide can only be identified by optical satellite remote sensing technology.

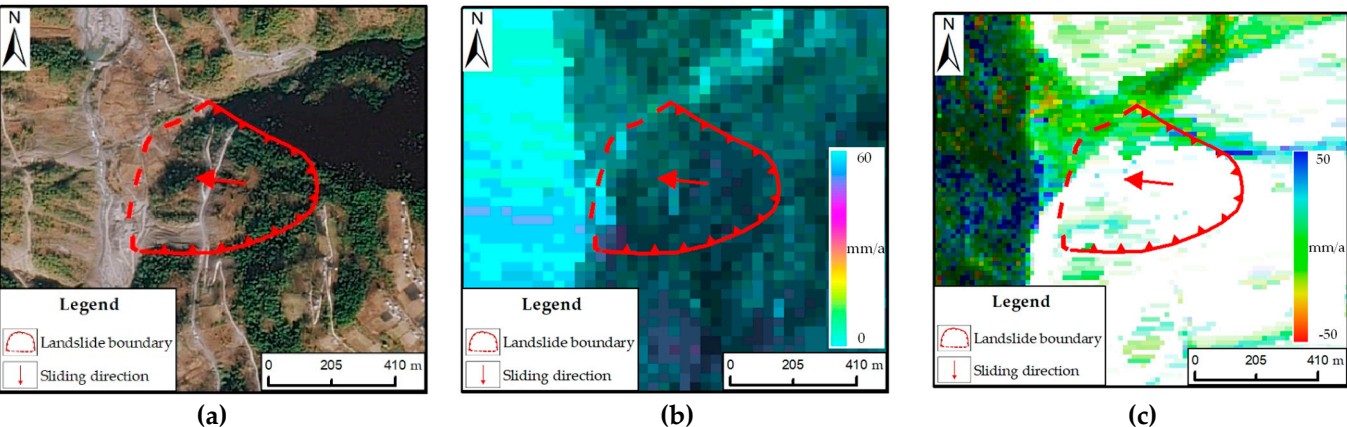

**Figure 8.** GSHP005 landslide ((**a**) optical satellite image; (**b**) InSAR deformation map of Sentinel-1A data; (**c**) InSAR deformation map of ALOS-2 data).

We conducted an on-site investigation of this landslide and found that there were tensile cracks that had developed on the gentle slope platform at the trailing edge of the landslide, with a crack width of approximately 30–50 cm, and the extension direction was perpendicular to the slope aspect. The landslide is now unstable, which affects the passage of the S318 provincial road. If the slope slides down as a whole, it may block the Dimaluo River (Figure 9). The on-site investigation showed that it was indeed an active landslide, and the method correctly identified the landslide.

Taking the landslide YPHP005 in Yongping County, in the southern region, as an example, we specifically analyzed the reasons for the low identification rate of active landslides based on InSAR technology in the southern region. The landslide is located on the inner side of the road. From the optical satellite image, signs of recent slippage can be seen in the middle and lower part of the slope, and the overall deformation characteristics of the landslide are obvious (Figure 10a). Due to the overall dense vegetation on the slope and the small scale of the landslide, the InSAR results of Sentinel-1A and ALOS-2 data showed no obvious deformation zones (Figure 10b,c). Therefore, the landslide can only be identified by optical satellite remote sensing technology.

We also conducted a field investigation of this landslide. The landslide caused a further small-scale landslide due to the construction of the road and the excavation of the slope toe. The recent superficial slide can be seen on the middle and lower part of the landslide. It mainly affects the G320 national road at the foot of the slope (Figure 11). The field investigation showed that it was indeed an active landslide, and the method correctly identified the landslide.

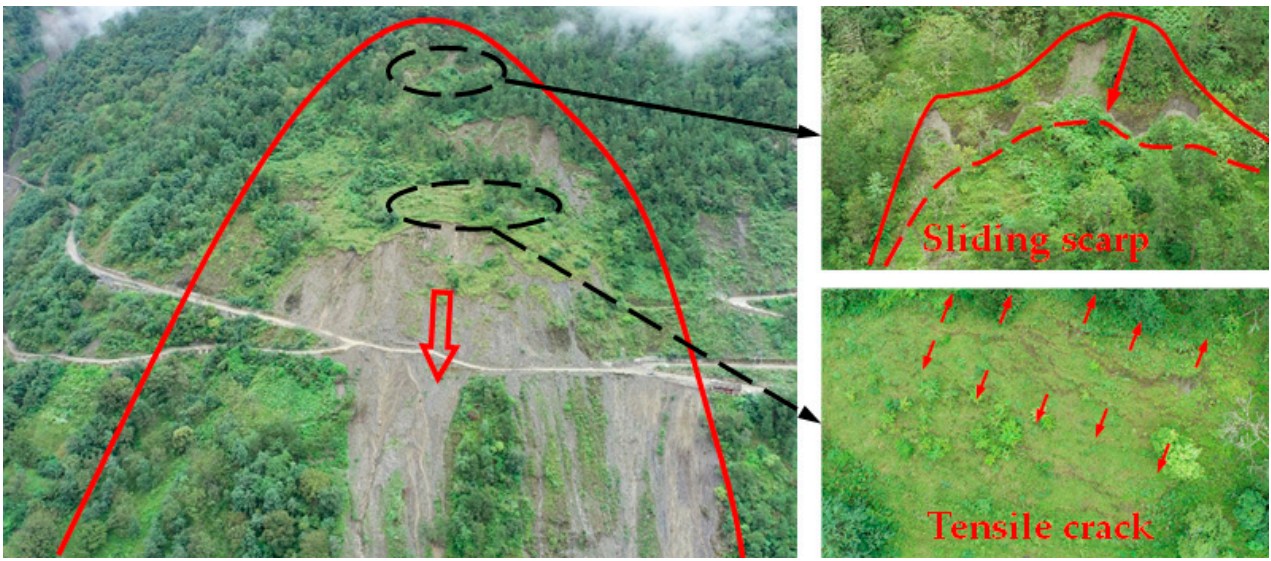

**Figure 9.** Photos of the GSHP005 landslide.

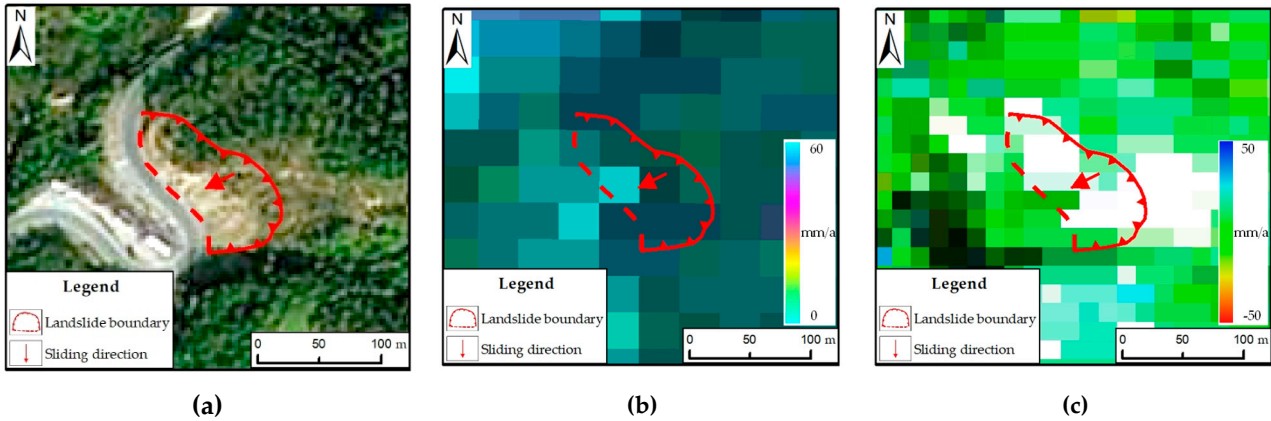

**(a)**　　　　　　　　　　　　**(b)**　　　　　　　　　　　　**(c)**

**Figure 10.** YPHP005 landslide ((**a**) optical satellite image; (**b**) InSAR deformation map of Sentinel-1A data; (**c**) InSAR deformation map of ALOS-2 data).

Therefore, this study shows that the terrain height and slope of the slope have a certain influence on the InSAR identification efficiency of landslides. In the high mountain and valley areas, with excessive terrain height and slope, a large area of SAR imaging contains large shadows and layover areas due to the large fluctuations in terrain and dense clouds and fog. Therefore, the effective identification rate of InSAR will be reduced. In the gentle hilly areas, with a small terrain height and slope, due to the geological environmental conditions, the scale of landslides is mostly small and medium, and the small size of the landslides will also affect the effective identification by InSAR. In the area of medium terrain height and moderate slope, the InSAR identification efficiency for active landslides is best. In the central region with moderate terrain height and slope, shadows and overlays do not exist in a large area, and the scale of landslides is generally not too small, so the InSAR identification efficiency of active landslides in this region is highest.

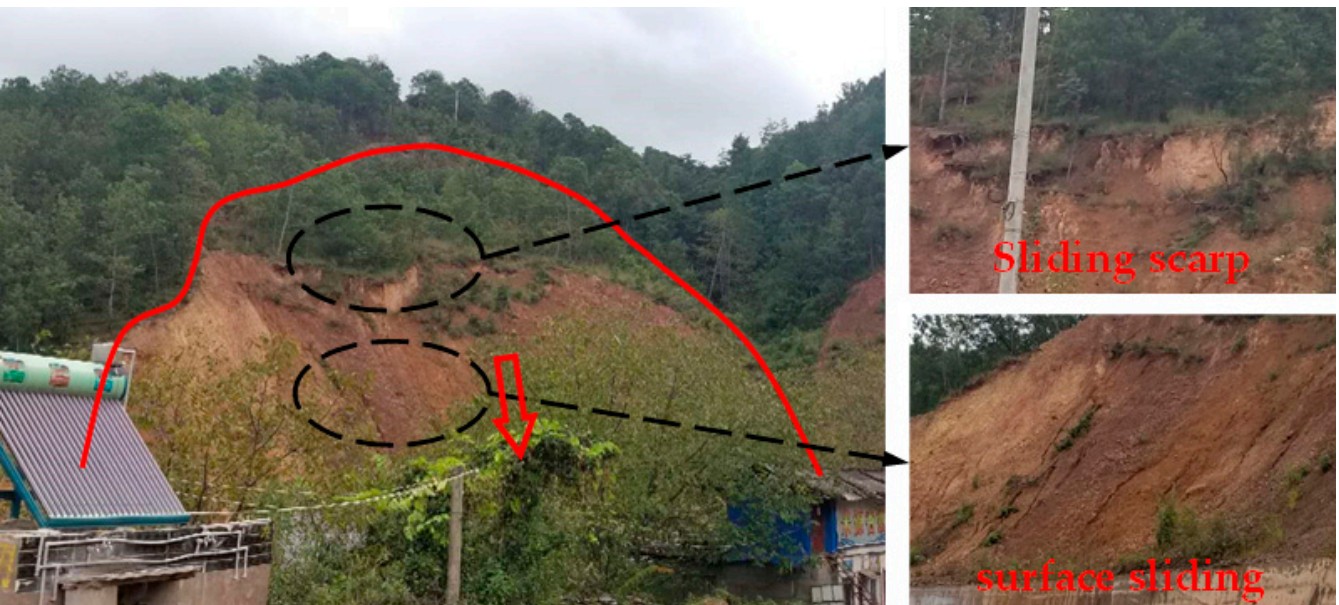

**Figure 11.** Photos of the YPHP005 landslide.

*5.2. Comparison of Landslide Identification Effect between ALOS-2 Data and Sentinel-1A Data by InSAR*

In addition, we also conducted a comparative analysis of the active landslide identification effects between Sentinel-1A and ALOS-2 data. As mentioned above, the active landslide identification efficiency $\eta_{\text{InSAR-S}}$ based on Sentinel-1A data and the active landslide identification efficiency $\eta_{\text{InSAR-A}}$ based on ALOS-2 data can be expressed as:

$$\eta_{\text{InSAR-S}} = \frac{n_{1\text{-}S} + n_{2\text{-}S}}{n_1 + n_2 + n_3} \tag{2}$$

$$\eta_{\text{InSAR-A}} = \frac{n_{1\text{-}A} + n_{2\text{-}A}}{n_1 + n_2 + n_3} \tag{3}$$

In Equation (3), $\eta_{\text{InSAR-S}}$ is the identification efficiency of active landslides based on Sentinel-1A data; $n_{1\text{-}S}$ is the number of active landslides with abnormal surface deformation values based on Sentinel-1A data but without obvious optical deformation characteristics; $n_{2\text{-}S}$ is the number of active landslides with abnormal surface deformation based on Sentinel-1A data, which also have obvious optical deformation characteristics; $\eta_{\text{InSAR-A}}$ is the identification efficiency of active landslides based on ALOS-2 data; $n_{1\text{-}A}$ is the number of active landslides with abnormal surface deformation values based on ALOS-2 data but without obvious optical deformation characteristics; $n_{2\text{-}A}$ is the number of active landslides with abnormal surface deformation based on ALOS-2 data, which also have obvious optical deformation characteristics.

The statistical results are shown in Figure 12 from the northern region to the southern region; the overall trends of $\eta_{\text{InSAR-S}}$ and $\eta_{\text{InSAR-A}}$ are essentially the same as those of $\eta_{\text{InSAR}}$, showing a trend of first increasing and then decreasing. $\eta_{\text{InSAR-S}}$ and $\eta_{\text{InSAR-A}}$ in the central region are higher than those in the northern and southern regions, respectively. In addition, the $\eta_{\text{InSAR-S}}$ of both the northern and central regions is higher than the $\eta_{\text{InSAR-A}}$ of the respective regions, indicating that, in the northern and central regions, more active landslides were identified based on Sentinel-1 data than those identified based on ALOS-2 data.

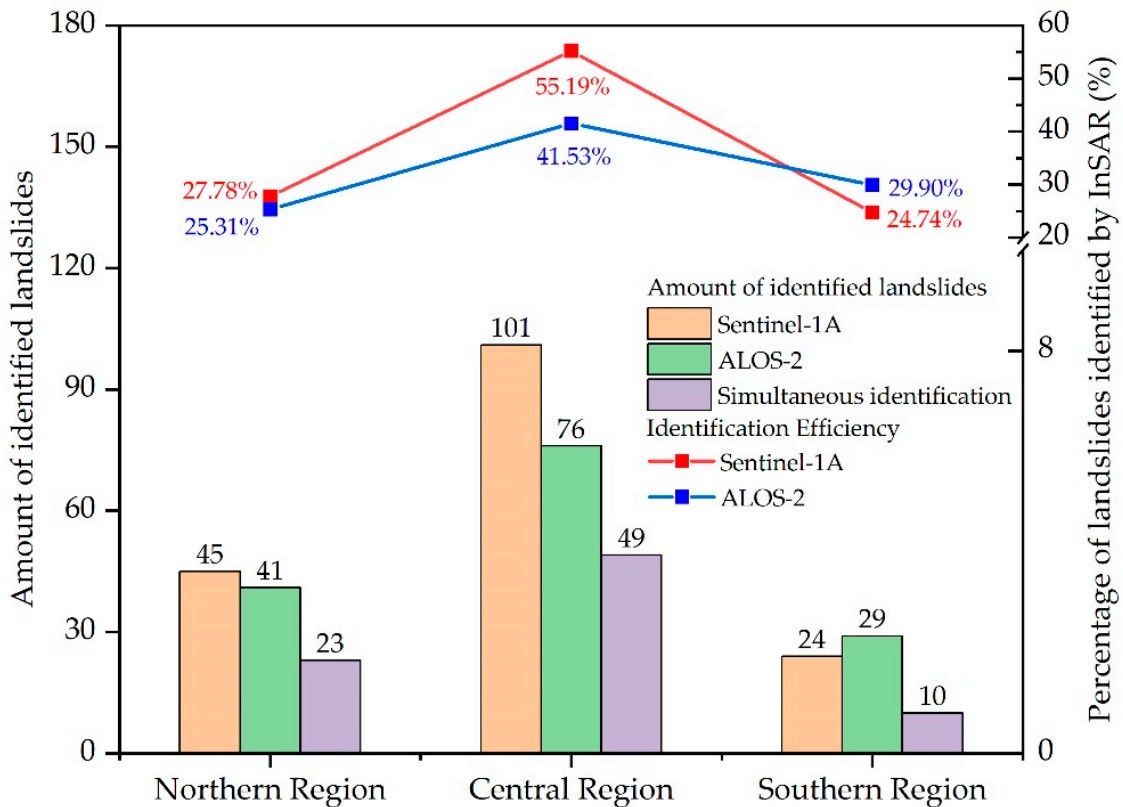

**Figure 12.** Comparison of InSAR identification effects with different data sources.

Let us take the LSHP012 landslide in Lushui City, Central District as an example. The landslide is located on the right bank of the Nu River. The landslide is approximately 1100 m long and 550 m wide, indicating that it is a large-scale landslide. On the optical satellite image, no obvious large-scale deformation characteristic is observed (Figure 13a). The InSAR results based on Sentinel-1A data show that the location of the middle and lower settlements of the landslide body has a large range of deformation zones (Figure 13b), while the InSAR results based on ALOS-2 data show no obvious deformation zone (Figure 13c).

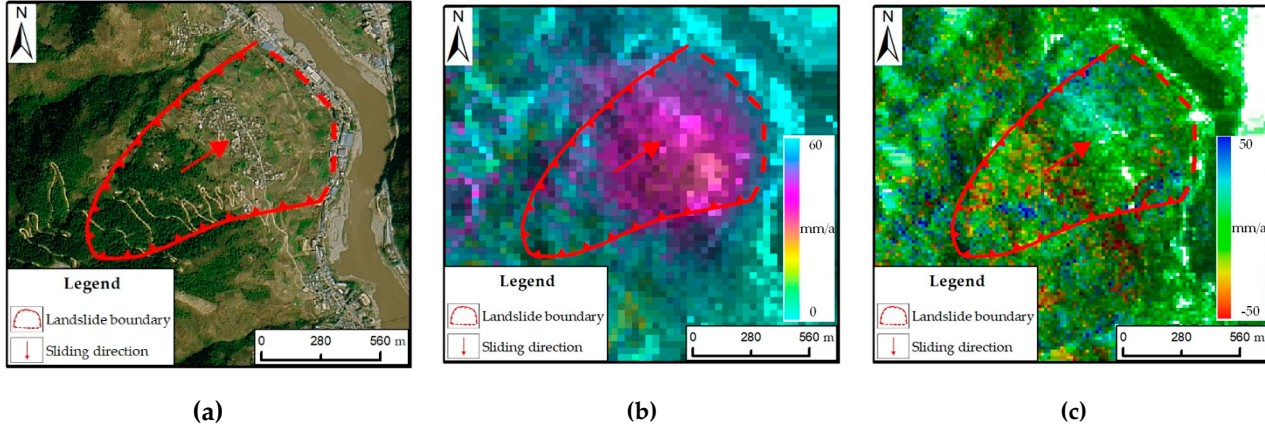

**(a)**        **(b)**        **(c)**

**Figure 13.** LSHP012 landslide ((**a**) optical satellite image; (**b**) InSAR deformation map of Sentinel-1A data; (**c**) InSAR deformation map of ALOS-2 data).

According to the on-site investigation of the landslide, there is a highway under construction in the middle of the slope, and the deformation is mainly concentrated in the settlement area in the middle of the slope. The distribution of deformation characteristics such as large-scale tensile cracks can be seen in many houses and retaining walls (Figure 14).

This location is indeed an active landslide with strong deformation, and the active landslide is identified by the InSAR results based on Sentinel-1A data. The Sentinel-1 data have more periods than the ALOS-2 data in this study. The Sentinel-1 data's revisit period is short and has strong coherence. Due to the long ALOS-2 data period and the large time interval, which results in poor coherence, the LSHP012 landslide cannot be identified.

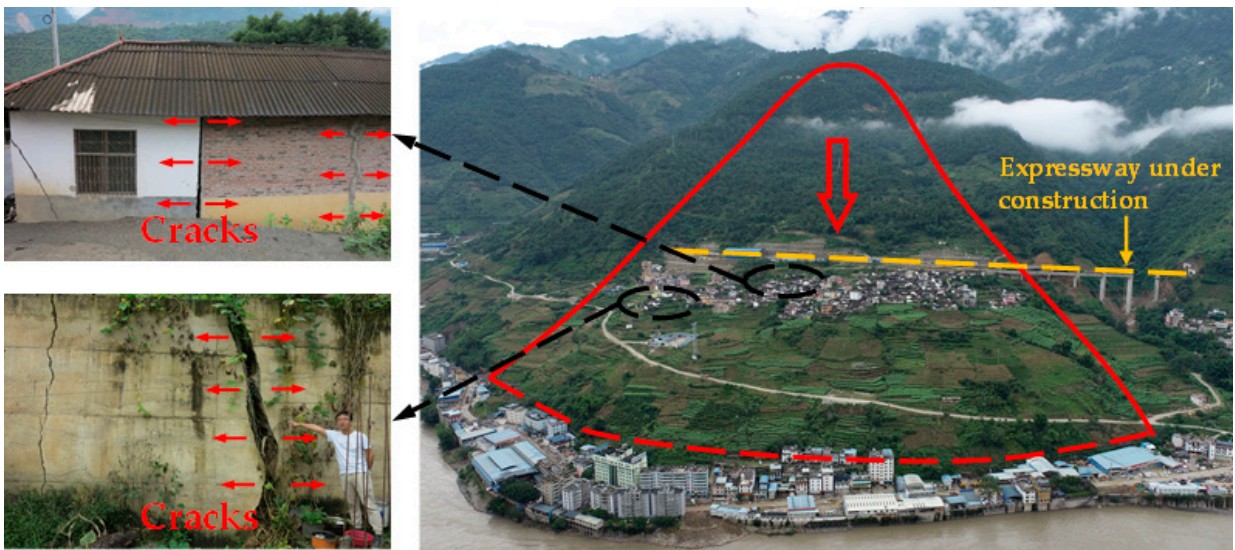

**Figure 14.** Photos of the LSHP012 landslide.

In addition, according to statistics, $\eta_{\text{InSAR-S}}$ in the southern region is lower than $\eta_{\text{InSAR-A}}$, and the number of active landslides identified based on Sentinel-1 data is less than that identified based on ALOS-2 data.

Let us take the YPHP008 landslide in Yongping County, in the southern region, as an example. The landslide is around 200 m in length and 450 m in width, and is a medium-scale landslide. On the optical satellite image, the vegetation on the left side of the landslide is dense, and no obvious deformation characteristics are found, while the vegetation on the right side of the slope is sparse, showing obvious signs of sliding (Figure 15a). The InSAR results based on Sentinel-1A data showed no obvious distribution of deformation zones (Figure 15b), while the InSAR results based on ALOS-2 data showed that large-scale deformation zones were distributed in the middle of the slope (Figure 15c).

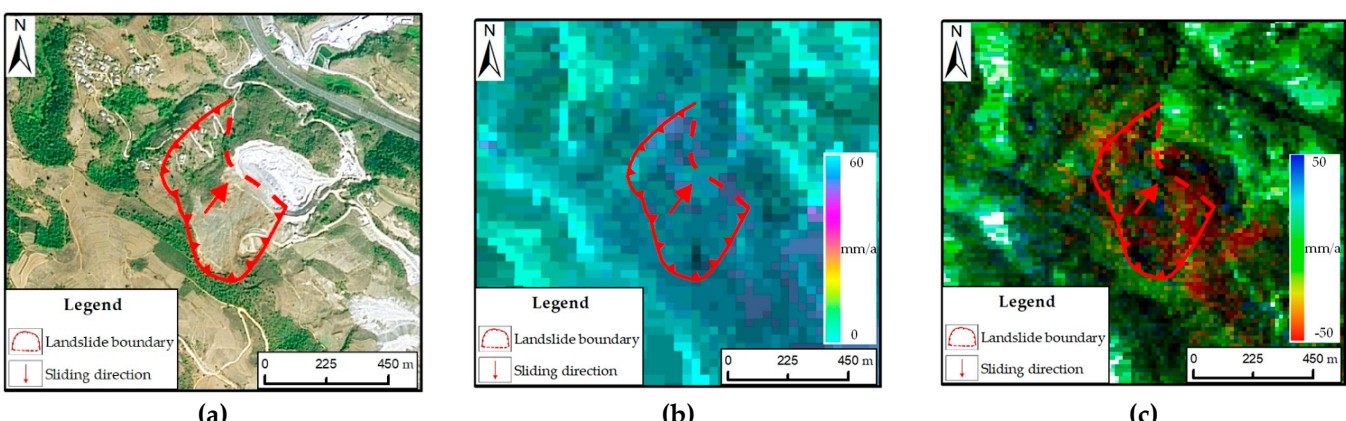

**Figure 15.** YPHP008 landslide ((**a**) optical satellite image; (**b**) InSAR deformation map of Sentinel-1A data; (**c**) InSAR deformation map of ALOS-2 data).

According to the on-site investigation of the landslide, the front edge of the landslide is an open-pit mining area (at the time of writing, the mining activities have stopped). Due to

the large-scale excavation of the front edge of the slope in the early stage, the deformation of the landslide was caused. On the left trailing edge of the landslide, a newly descended steep ridge with a height of approximately 5 m can be seen, and the cracks on the surface of the slope are densely distributed, causing some houses to collapse (Figure 16). This landslide belongs to the active landslides induced by mining, and the landslide deformation is relatively strong. The identification result based on the ALOS-2 data is correct. Due to the small scale of the landslide and the dense vegetation on the left side of the slope, the Sentinel-1A data, with low resolution and weak vegetation penetration, cannot identify the YPHP008 landslide.

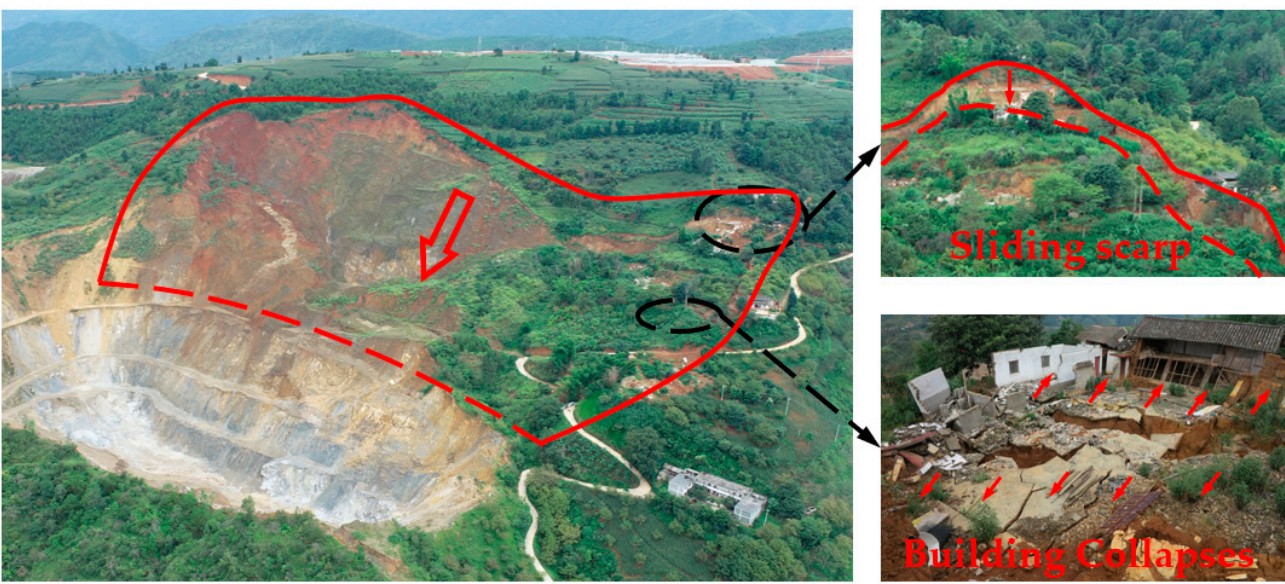

**Figure 16.** YPHP008 landslide scene photos.

In the field of identifying active landslides, Sentinel-1A data and ALOS-2 data have their own advantages. Sentinel-1A data have a short revisit period and the data are free. Therefore, a large amount of Sentinel data are beneficial in identifying large-scale and clearly deformed active landslides. The advantage of ALOS-2 data is that the data have strong penetrability regarding the vegetation and high resolution, which can effectively identify small- and medium-scale landslides in high-density vegetation coverage areas. Therefore, we suggest that the Sentinel-1A data and ALOS-2 data should be used together in the process of active landslide identification in large areas, which is beneficial to improve the identification efficiency of active landslides using InSAR technology.

*5.3. Comparison of InSAR and Optical Satellite Remote Sensing*

When using InSAR technology to identify active landslides in high mountains and valleys, problems such as shadows and layovers are unavoidable [19], which can lead to the missed identification of active landslides [35]. To solve this problem, scholars have proposed that the orbit-ascending and orbit-descending data of SAR should be used in combination [22]. In addition to using the Sentinel-1A data and ALOS-2 data in combination, as proposed above, the combination of optical satellite remote sensing and InSAR technology should also be strengthened.

On the latest optical satellite images, if there are suitable terrain conditions and obvious deformation characteristics, the location is recommended to be identified as an active landslide. As mentioned above, the GSHP005 landslide and the YPHP005 landslide failed to be identified by InSAR due to factors such as strong terrain cutting, dense vegetation, and a small landslide scale. However, based on optical satellite images, the two landslides were identified as active landslides with suitable topographical conditions and different

levels of deformation characteristics. Evidence from field investigations proves that these two sites are indeed active landslides.

Figure 17 shows the statistical results of the number of active landslides identified based on optical satellite remote sensing technology. Only about 50–75% of the active landslides in the TPRR can be identified by optical satellite remote sensing technology. Therefore, in order to identify all the active landslides in the TPRR accurately, InSAR technology and the optical remote sensing technology identification method should be combined. This method can effectively make up for the missed identification of active landslides caused by the shadow and overlay of InSAR technology, and can greatly improve the accuracy of large-area active landslides reservoir maps.

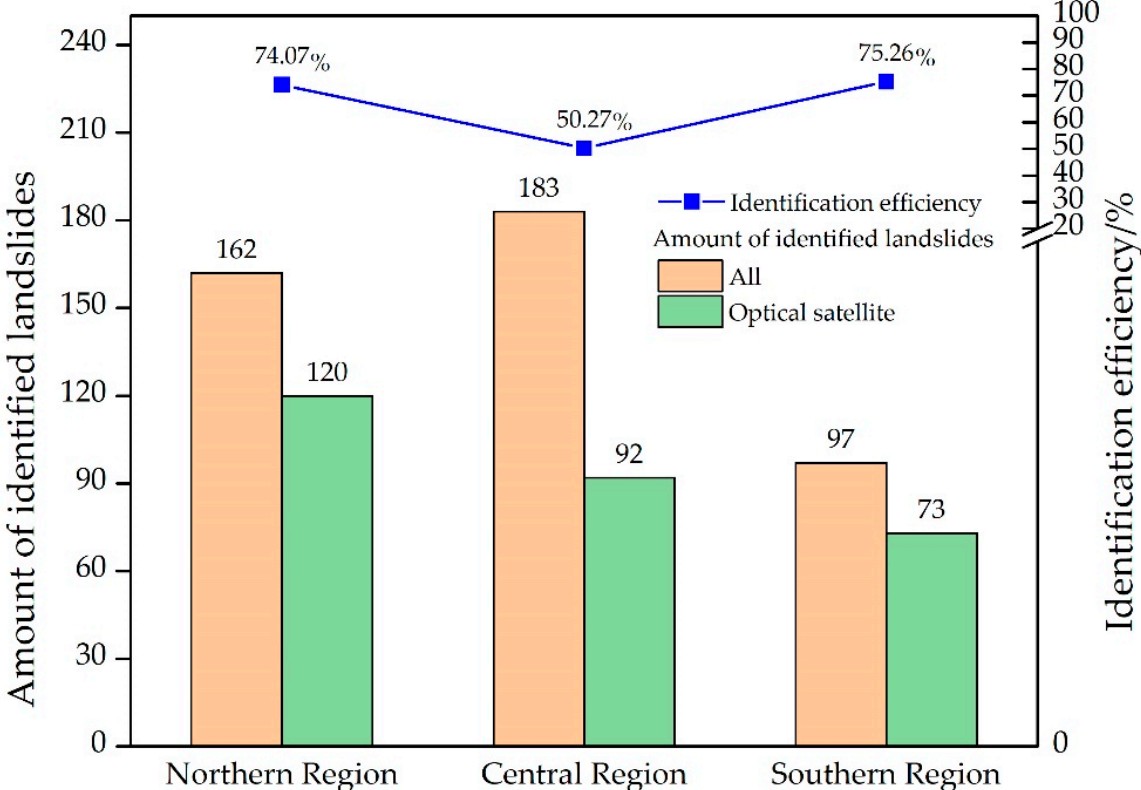

**Figure 17.** Statistics of the number of active landslides identified based on optical satellites.

Therefore, we believe that the combined identification of active landslides with InSAR technology and optical remote sensing can effectively compensate for the missed identification of active landslides caused by shadowing and layovers in InSAR technology, and can greatly improve the accuracy of the active landslide inventory maps in large areas.

### 5.4. Distribution of Landslides and Fault Zones

Active fault zones have an important impact on the distribution of geohazards such as landslides [36]. There are many active fault zones in the TPRR on the southeastern margin of the Qinghai–Tibet Plateau.

In order to analyze the relationship between landslides and fault tectonic belts in this area, we divide the distances between landslides of different scales and fault tectonic belts in the TPRR into four categories: 0–200 m, 200–500 m, 500–1000 m, >1000 m. The closer the landslide is to the fault zone, the closer the relationship between the landslide and the fault zone. The zone data in the TPRR in this study are sourced from the 1:250,000 regional geological survey in this area.

According to the field investigation results, among the 442 landslides identified in this study, 85 are small-scale landslides; 137 are medium-scale landslides; 155 are large-scale

landslides; 60 are super-large-scale landslides; 5 are huge-scale landslides. The distances between landslides of various scales and fault zones are shown in Figure 18.

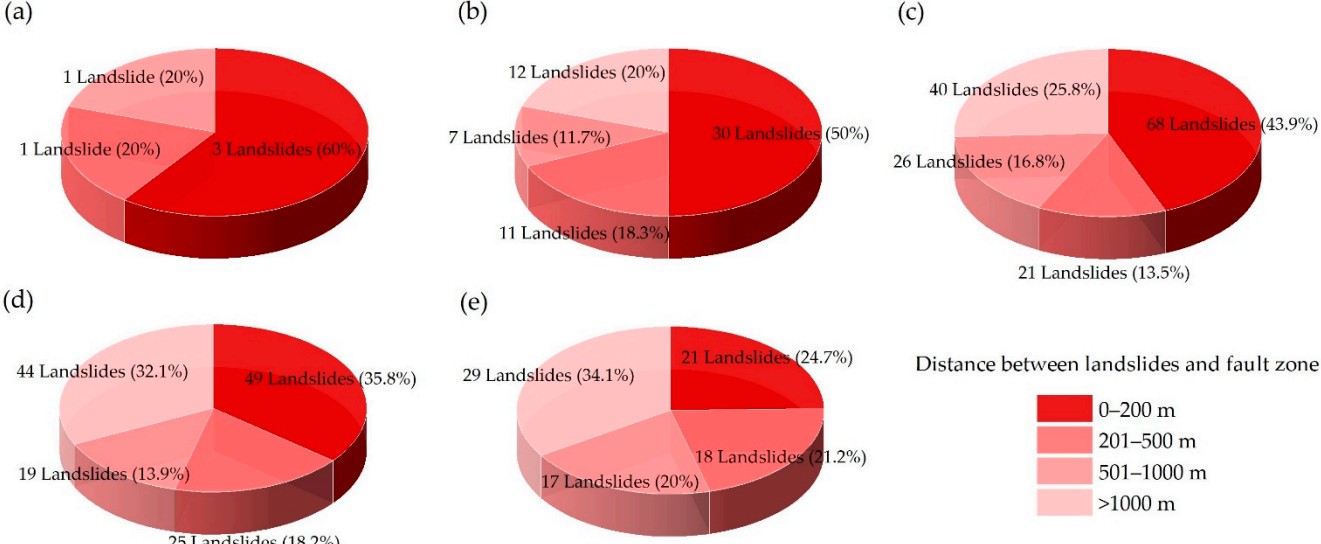

**Figure 18.** The distance between the active landslide and the fault zone in the TPRR ((**a**) the distance between the huge-scale landslide and the fault zone; (**b**) the distance between the super-large-scale landslide and the fault zone; (**c**) the distance between the large-scale landslide and the fault zone; (**d**) the distance between the medium-scale landslide and the fault zone; (**e**) the distance between the small-scale landslide and the fault zone).

All the huge landslides in the TPRR are located within 1000 m on both sides of the fault zone, and 60% of the huge landslides are located within the range of 200 m on both sides of the fault zone. Moreover, 50% of the super-large landslides are located within the range of 200 m on both sides of the fault zone. When the scale of the landslide is smaller, the proportion of landslides on both sides of the fault zone is smaller. Only 24.7% of the small landslides are located within the range of 0 to 200 m on both sides of the fault zone, and 65.9% of the small landslides are located within the range of 1000 m on both sides of the fault zone.

The formation and development of landslides are determined by many factors, such as topography, stratigraphic lithology, fault zones, human engineering activities, and rainfall [37], among which the fault zone is one of the more important factors. The larger the landslide scale, the greater the impact of the fault structure on the landslide. The fault structure provides a broken rock mass. The closer the landslide is to the fault structure, the more fragmented the rock mass, and the more developed the joints, cleavage, and mylonite of the rock mass; thus, it is easier for large-scale landslides to develop. For small-scale landslides, such as the YPHP005 landslide mentioned above, due to the construction of the road and the excavation of the slope foot, a free surface is formed; under the condition of heavy rainfall, the formation of small-scale landslides is induced, and the fault structure was not the necessary factor for landslides formation and development.

## 6. Conclusions

In this paper, an active landslide identification method combining InSAR technology and optical satellite remote sensing technology is proposed, and the method is successfully applied to the TPRR in the northwest of Yunnan Province, China. In this area of approximately 65,800 km$^2$, a total of 442 active landslides were identified, and the overall identification rate of active landslides based on InSAR technology reached 51.36%.

The area of the TPRR is vast, and the altitude and slope of the mountain gradually decrease from north to south. The altitude and slope not only affect the formation and de-

velopment of landslides, but they also affect the identification efficiency of active landslides using InSAR technology. In the northern region, with steep terrain and large terrain cuts, problems such as shadows and overlays can easily lead to InSAR failure; in the southern region, with gentle terrain, the scale of landslides is generally small, which is not suitable for the application of InSAR technology. In the central region, with moderate terrain and slope, the identification rate of active landslides based on InSAR technology is highest, reaching 66.12%.

Sentinel-1A data and ALOS-2 data have their own advantages, and the combined use of the two data sources is beneficial for improving the identification rate of active landslides using InSAR technology. The combined active landslide identification method can reduce the missed identification of landslides and help to improve the accuracy of large-area active landslide inventory maps. This active landslide identification method can be applied to wide-area landslide identification.

The TPRR has complex geological conditions and extremely developed fault structures. The distribution of active landslides in this area is closely related to the fault structural belt. Overall, 70% of the active landslides are distributed within 1 km on both sides of the fault structural belt. The larger the scale of the landslide, the closer the relationship between the landslide and the fault structure and the higher the proportion of landslides close to the fault structure belt. In the process of landslide identification in this area, attention should be paid to the influence of active fault structures.

**Author Contributions:** Conceptualization, C.Z. and J.L.; data curation, C.Z., J.D., and S.Z.; investigation, J.L., C.Z., S.Y., L.Y., and B.L.; methodology, C.Z. and J.L.; writing—original draft preparation, S.Z. and X.M.; writing—review and editing, C.Z., J.L., and W.L. All authors have read and agreed to the published version of the manuscript.

**Funding:** This work was funded by ministry of natural resources national geological hazard identification project in high risk areas (No. 0733-20180876).

**Institutional Review Board Statement:** Not applicable.

**Informed Consent Statement:** Not applicable.

**Data Availability Statement:** The Sentinel-1A data in this study were provided by the European Space Agency (ESA) through the Sentinel-1 Scientific Data Hub. The ALOS-2 data and GF-1 datasets were provided by the Department of Geological Exploration Management, Ministry of Natural Resources, China.

**Acknowledgments:** We are very grateful for the Sentinel-1A data provided by the European Space Agency (ESA), and the ALOS-2 data and GF-1 datasets provided by the Department of Geological Exploration Management, Ministry of Natural Resources, China.

**Conflicts of Interest:** The authors declare no conflict of interest.

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
