# Peer review of "Integration of Sentinel-1A, ALOS-2 and GF-1 Datasets for Identifying Landslides in the Three Parallel Rivers Region, China"

_remotesensing, doi:10.3390/rs14195031_

Round 1

Reviewer 1 Report

Dear authors,

thank you for the manuscript. It regards landslide detection based on Sentinel-1 InSAR, ALOS-2 InSAR and optical data analysis in a given area, and what seems most interesting to me, is the InSAR landslide identification rate for each area.

The identification rate is evaluated separately for each area. On the other hand, I think that the identification rate should be evaluated independently (all areas can be evaluated together, on the other hand) for different landcover, landslide velocities, sensitivity with regard to the satellite track (i.e. slope and its orientation, ideally numerically) and size of the landslide (w.r.t. the SAR resolution / spatial filter).

InSAR processing: did you do spatial filtering? how strong filter was used? do you identify  incoherent areas?

Minor comments:

- please use upper /lower indices in text, such as m^2 -> m2

- line 112: middle area -> central area

- section 3.1.2: please give reasons for different period of optical and SAR images

- line 299: slope of the slope

- please use "layover" or "overlay" instead of "overlapping" (more occurences)

- fig 10: please include colorscales

- line 306-308: "in the area of medium terrain height..." I consider this sentence too generalizing, the identification rate is supposedly not dependent on the height

Reviewer 2 Report

Fig 2 does not show landslides, even though they are in map explanations

FIg 6 - the overviews in the right part of the figure do not bring any new information, they are useless

3.2.2

There is no information on how landslides were detected on the optical data. So were the obvious optical features searched manually or was some automatic or semi-automatic method used?

Ad discussion, 5.1:

I believe that the division of the regions into northern, central and southern is not appropriate. There are high mountain ranges with a large slope in both the North and Central regions, and they are also found in the South region. Even in terms of further statistical evaluation of the success of the methods, the distribution is not meaningful. It would be better to group landslides only by slope gradient and not to define regions. Statistically processing these groups (by formula 1), the results for these groups would be more informative.

It would then become apparent that it is not the north, south or central region that plays a role in the success of InSAR, but the slope gradient, its orientation (due to the position of the SAR antenna) and the ruggedness of the terrain - as you write very well in paragraph line 299-307.

Ad 5.2

Same comment on the division of regions.

It would be much more interesting to evaluate the success of methods by slope, orientation, etc. It would also be worthwhile to separate the landslides detected in the satellite ascending and descending path images. The limits of the data and methods used would then be better highlighted.

Furthermore, the 3 columns of Insar, Sentinel-1 and ALOS-2 in Figure 12 are a bit misleading. Because the Insar column is actually both Sentinel-1 and ALOS-2. It would be better to see how many landslides are only S-1, how many are only ALOS-2 and how many were detected simultaneously by both satellites (composite bar chart).

Again, the best verbal assessment is line 380-388, which better reflects reality than FIg 12.

Line 358 - start new paragraph

Ad 5.3

There is no statistical evaluation here, unlike in Sections 5.1 and 5.2. Why? If the chapter is listed here, there should be information on what % of the landslides were visible only on the optical data. It is not clear from the text if there were only two landslides GSHP005 and YPHP005 or if there were more and if there were other causes than those listed in lines 400 and 401.

Reviewer 3 Report

This paper presents a case study of landslide detection in TPRR area using SAR and optical images. The authors have done a lot of work which have potentials to be published in Remote Sensing journal. However the readability of the paper is extremely low with a lot of repeated information or useless expressions. The contents and structures of the manuscript should be carefully recognized and improved. The results presented are also inadequate. Issues should be addressed are listed as follows. (1)  Extensive editing of English language and style are required. (2)  The abstract of the article is inadequate. Therefore, a summary should be written to better reflect the work from start to finish. (3)  I think “steep terrain, dense vegetation, and complex clouds” are natural phenomena. In the meantime, microwaves can penetrate clouds. There are a lot of similar inaccurate expressions in the manuscript. (4)  The introduction is quite disorder and confusion. Advantages of InSAR and optical remote sensing should be summarized. Why the authors chose to combine InSAR and optical remote sensing should be given. (5)  Line 60-62, it seems Wang et al. used ALOS PALSAR and Sentinel-1. (6)  The author claims they proposed a method combine InSAR and optical remote sensing. What is new in your methods compared to previous similar methods should also be compared. (7)  Line 101-102, the location of “the Nujiang river fault zone, the Lancang river fault zone and the Jinsha river fault zone” should be given in Fig.1. (8)  Line 113-123, this paragraph should be recognized. (9)  Section 3.1, the data used in this study are SAR data which is total different with radar data. The Sentinel-1 is an SAR sensor. (10)             The resolution of ALOS2 given in the main text and table 1 are different. Pls check. (11)             List of differential InSAR pairs used for stacking InSAR can be given. (12)             Coverage of GF-1 images should also be marked in Fig. 2. (13)             Line 160-166, these texts are well-known knowledge which can be removed or given in the introduction. (14)             Section 3.2.2, how do the authors extract the deformation characteristics and landslide boundaries from the optical images should be addressed. (15)             The landslide boundaries from InSAR and optical remote sensing are generally different? What method does the author use to extract the boundaries? (16)             Line 208-211, there are a lot of impact factors for sliding. Why the authors conclude hydropower stations impact the development of landslides? (17)             Fig. 6, what is the Sentinel-1 and ALOS2 represent? If they are deformation rate maps, color bars should be given. Why the colors between Sentinel-1 and ALOS2 are different? (18)             Section 5, the structures should be recognized and improved in depth. (19)             The authors make me confused. The authors claimed they used ascending and descending Sentinel-1 datasets. Measurements from ascending and descending datasets are expected to be totally different. However, Fig. 6, 8, 10, 13 and 15 only give one Sentinel-1 results. How do the authors combine the results from different orbit? (20)             Impact factors of landslide should be discussed.

Round 2

Reviewer 3 Report

The authors have answered my concerns.